# Influence of Mechanical Fatigue at Different States of Charge on Pouch-Type Li-Ion Batteries

**DOI:** 10.3390/ma15165557

**Published:** 2022-08-12

**Authors:** Jin-Yeong Kim, Jae-Yeon Kim, Yu-Jin Kim, Jaeheon Lee, Kwon-Koo Cho, Jae-Hun Kim, Jai-Won Byeon

**Affiliations:** 1Department of Materials Science and Engineering, Seoul National University of Science and Technology, Seoul 01811, Korea; 2Department of Mining and Geological Engineering, University of Arizona, Tucson, AZ 85721, USA; 3Department of Materials Engineering and Convergence Technology, Research Institute for Green Energy Convergence Technology, Gyeongsang National University, 501, Jinju-daero, Jinju-si 52828, Korea; 4School of Materials Science and Engineering, Kookmin University, Seoul 02707, Korea

**Keywords:** Li-ion battery, flexible battery, mechanical fatigue test, state of charge, electrochemical impedance spectroscopy

## Abstract

Since flexible devices are being used in various states of charge (SoCs), it is important to investigate SoCs that are durable against external mechanical deformations. In this study, the effects of a mechanical fatigue test under various initial SoCs of batteries were investigated. More specifically, ultrathin pouch-type Li-ion polymer batteries with different initial SoCs were subjected to repeated torsional stress and then galvanostatically cycled 200 times. The cycle performance of the cells after the mechanical test was compared to investigate the effect of the initial SoCs. Electrochemical impedance spectroscopy was employed to analyze the interfacial resistance changes of the anode and cathode in the cycled cells. When the initial SoC was at 70% before mechanical deformation, both electrodes well maintained their initial state during the mechanical fatigue test and the cell capacity was well retained during the cycling test. This indicates that the cells could well endure mechanical fatigue stress when both electrodes had moderate lithiation states. With initial SoCs at 0% and 100%, the batteries subjected to the mechanical test exhibited relatively drastic capacity fading. This indicates that the cells are vulnerable to mechanical fatigue stress when both electrodes have high lithiation states. Furthermore, it is noted that the stress accumulated inside the batteries caused by mechanical fatigue can act as an accelerated degradation factor during cycling.

## 1. Introduction

Nowadays, flexible energy storage devices are in strong demand because of the rapid development of flexible electronics and mobile wearable devices. Power sources play an important role in the use of flexible devices and much research effort has been dedicated to this area. Among the various energy storage devices, Li-ion batteries (LIBs) are considered one of the most suitable devices for flexible electronic applications due to their high energy density, long cycle life, and reasonable cost [1]. For commercial LIBs, layered transition metal oxides (LiCoO_2_ (LCO), LiNi_x_Co_y_Mn_z_O_2_ (NCM), x + y + z = 1) have been widely used as cathode materials, thanks to their electrochemical properties and cost [2,3]. As for the anode, graphite has generally been adopted due to its good cycle performance and moderate capacity [4,5]. To apply the LIBs to flexible applications, the durability of the LIBs against mechanical fatigue stress is one of their essential properties. Moreover, the degradation behavior of mechanically fatigued LIBs should be investigated to facilitate the new development of mechanically reliable LIBs and their proper usage under mechanical fatigue.

General degradation mechanisms of LIBs have been investigated in many previous studies. Kim et al. presented the effect of the LiF-based solid electrolyte interphase (SEI) layer on the surface of graphite anodes [6]. It was revealed that the semi-C-F bonds promoted the formation of stable LiF-based SEI layers. Uhlmann et al. examined the Li plating mechanism on graphite by the variation in charging current based on the Butler–Volmer equation [7]. The formation of the Li metal on anode surfaces during the Li insertion process into anode materials could result in serious capacity fade [8,9]. In addition, the dendrite growth of Li metal could cause safety problems by short-circuiting cells [10,11,12,13]. Zhang et al. investigated the microstructural changes of layered oxide/graphite cells during a cycling test [14]. Some cracks were observed on the two electrode particles.

The main degradation mechanisms of mechanically fatigued LIBs were reported to be electrode fracture [15], delamination between current collectors and active materials [16,17,18], and cracking of active layers on both electrodes [17,19,20]. The electrode consists of a current collector and an active layer. The active layer on the current collector is normally a mixture of lithium transition metal oxide (e.g., LCO, NCM, and so on), high conductivity carbon, and a binder. The active layer is known to be brittle because it is physically mixed [21]. Therefore, when the LIBs are subjected to mechanical fatigue stress, the electrochemical contacts between current collectors and active materials, and each active particle could be weakened. Breaking a continuous network for electron transport can cause severe capacity loss [16,17,19].

Meanwhile, the phase transition and accordance volume expansion of active materials occur depending on the state of charge (SoC) of the battery. For instance, during the lithiation and delithiation, LCO undergoes some phase transitions, such as a metal–insulator phase transition in the low-voltage region (i.e., around 3.95 V) and an order/disorder transition at approximately 4.2 V from the hexagonal structure to a monoclinic structure [22,23,24]. Correspondingly, the unit cell volume of LCO is known to be changed to about 3~5% [23]. In the case of the NCM, its unit cell volume is contracted up to ~8.4% by the phase transition from the initial hexagonal phase (H1) to the monoclinic phase (M), second hexagonal phase (H2), and third hexagonal phase (H3) during the delithiation (i.e., charging) processes [25,26]. The volume of graphite, a common anode material, could be expanded to 13.2% when the graphite (C_6_) is fully lithiated to a composition of LiC_6_ [27]. In summary, the internal environment of LIBs in terms of the phase and volume of active materials are different depending on the SoC. Therefore, we believe that the effect of mechanical fatigue stress on the electrochemical characteristics of LIBs could be different depending on the SoC. However, there have been no reports on this topic, although the effect of different types of fatigue modes (e.g., bending, torsion, and complex strain) on the durability of LIBs has been investigated in many previous studies, including those by our group [15,16,17,18,19,20,21,28].

In this study, the effect of fatigue stress on the LIBs with different initial SoCs was investigated for the first time. Ultrathin pouch-type lithium-ion polymer batteries (Li-Pos) with bendability were used for the sake of the mechanical fatigue test, because a large number of flexible batteries are not commercially available, although various types of flexible LIBs have been developed at the research level [15,16,17,18,19,20,21]. The ultrathin pouch-type Li-Pos with different initial SoCs were subjected to repeated mechanical fatigue stress and then galvanostatically cycled 200 times. Since the main degradation mechanisms in the fatigued flexible batteries are delamination and cracking, leading to an increase in the contact resistance between the components, electrochemical impedance spectroscopy (EIS), which is capable of evaluating the internal electrochemical resistance of the battery, was used in this study. Moreover, a puncturing test was performed to investigate the effect of torsional stress on the separator, which is important for safety concerns with LIBs.

## 2. Materials and Methods

### 2.1. Sample Characterization

The ultrathin pouch-type Li-Pos with bendability (81 × 20 × 0.5 mm) were used for testing and were commercially provided by Powerstream Co. Ltd., Orem, UT, USA (model: PGEB0052081). This battery consists of a single cathode, anode, and separator. The full cells exhibited a 3.7 V nominal voltage and 15 mAh capacity. LiCoO_2_ and LiNi_x_Co_y_Mn_z_O_2_ materials were blended for the cathode (Appendix A and Appendix A), and graphite was used as the anode (Appendix A). Polypropylene separators were employed. There was a single cathode and anode.

### 2.2. Mechanical Fatigue Test

Figure 1 shows the experimental flow chart of this study. Before the mechanical fatigue test, charging and discharging of one cycle were performed on the tested cells and then the fully discharged batteries were charged to a specific voltage to set the designated SoC. The SoC at 100%, 70%, 60%, 40%, 20%, and 0% corresponded to 4.19 V, 3.82 V, 3.76 V, 3.66 V, 3.55 V, and 3.18 V of the batteries, respectively. The voltage points were calculated from the galvanostatic charge–discharge curves of the batteries at 0.3 C (6 mA, 1 C = 20 mAh). After setting the SoCs of the batteries, a mechanical test was conducted on each cell. A fatigue test was performed by using a micro-fatigue tester (E3000LT, Instron, High Wycombe, UK). To simulate the torsional and bending mode-like flexible condition, the fatigue test was conducted under sinusoidal loading. The *Y*-axis of the 0 point is S and the amplitude is A in this study. As shown in Figure 2a, for the torsion fatigue test, the phase change according to S, S + A, S, S − A, and S was defined as 1 cycle. Figure 2b shows the torsional fatigue test. Only the upper fixture performed the torsion and the lower fixture was fixed. Figure 2c shows the actual fatigue load changes based on the torsion of the cell. Torsional stress of 10° at a frequency of 1 Hz was applied to the batteries with different SoCs for 100,000 cycles. As shown in Figure 3a, for the bending fatigue test, the phase change according to S, S + A, and S was defined as 1 cycle. Figure 3b shows the bending fatigue test. The upper and lower fixture simultaneously performed the bending. Figure 3c shows the actual fatigue load changes based on the bending of the cell. A bending stress of 10, 15, and 20° at a frequency of 1 Hz was applied to the batteries with SoC 70% for 100,000 cycles. After the fatigue and cycling test, separators in the cells were subjected to a puncturing test, the process of which can be referred to in ASTM F1306. The test speed was 50 mm/min and the sample diameter was 34.9 mm.

### 2.3. Electrochemical Test

After the mechanical fatigue test (torsional test at 10°, 100,000 cycles), battery cycling was performed at constant current–constant voltage (CC–CV) for charging and at CC mode for discharging (WonATech Cycler, Seoul, Korea) at 25 °C. During the CC–CV mode for charging, a constant current of 6 mA was applied until the maximum voltage of 4.2 V was reached, followed by a constant voltage period until the current dropped below 0.1 mA (0.005 C). The discharge was performed at CC mode (6 mA) until the minimum voltage limit of 3.0 V, followed by a pause to stabilize the batteries. The charge and discharge processes were repeated 200 times. To investigate the phase transformation of the electrodes in the batteries, cyclic voltammetry (CV, WonATech WBCS3000, Seoul, Korea) was conducted for the voltage window of 3.0–4.2 V, with a scan rate of 0.01 mV/s.

### 2.4. Failure Analysis

The microstructure and elemental composition of the electrodes were analyzed by field emission scanning electron microscopy with an energy dispersive spectrometer (FE-SEM/EDS, Carl Zeiss, Jena, Germany). The FE-SEM was conducted under an accelerated voltage of 15 kV and a working distance of 8.5 mm using secondary image mode. Electrochemical impedance spectroscopy (EIS, Princeton Applied Research VersaSTAT4, Oak Ridge, TN, USA) was performed over the frequency range from 1 MHz to 0.01 Hz, with an amplitude of 10 mV. A software program was used to fit the spectrum (ZSimpwin, Warminster, PA, USA). For EIS analysis, the Li-Pos (initial SoC at 70%) were mechanically fatigued under two test modes (torsion and bending at various degrees for 100,000 cycles).

## 3. Results and Discussion

### 3.1. Cycle Performance after the Mechanical Test

To investigate the effect of mechanical fatigue on the cycle performance of the different SoC cells, the torsional fatigue tests were performed for up to 100,000 cycles prior to the charge/discharge cyclic test. The torsional angle was set at 10 degrees. The SoC indicates the charge state of the batteries prior to the mechanical fatigue test and, in this study, the cells were named after the initial SoC before the mechanical and cycling tests. For example, SoC 70% implies that the cell charged to 70% was subjected to the mechanical test. The capacity retention of the cells at each cycle is equivalent to the state of health (SoH = capacity after cyclic test/capacity at the initial cycle × 100). Figure 4a shows the open circuit voltage (OCV) of cells before and after the mechanical fatigue test. The OCVs of SoC 40–100% cells after the mechanical fatigue test were very slightly reduced. However, those of SoC 0% and 20% were reduced to 1.4 V and 1.9 V, respectively, which are values below the operating voltages (3.0 V~4.2 V) of this cell. This result implies that the SoC 0% and 20% cells physically failed due to the mechanical fatigue stress. Figure 4b,c shows the cycling performance of the fatigued cells, and the capacity values at the first, 20th, 40th, 60th, 80th, and 100th cycles are presented in Figure 4d. The electrochemical cycling tests were performed under the same conditions for all cells, and the voltage profiles at each cycle are presented in Appendix A. The capacity retention of all fatigued cells was more faded than that of the non-fatigued cell. Generally, the capacity fade of cells during the charge/discharge cyclic tests occurs due to electrochemical degradation phenomena, such as the accumulation of a solid electrolyte interphase (SEI) layer on the electrode surface, Li precipitation, and cracking of active materials [28,29]. According to some previous studies, it was reported that physical damages, such as the delamination between the current collector and active layer, cracking of the active layer, distortion of crystal atomic lattice of active materials and so on, could be induced to the electrode by mechanical fatigue stress and be accumulated inside the cell [15,16,17,18,19,20,21,28]. Thus, when the cell was subjected to a shear strain by a squeezing deformation-like torsion mechanical stress, the electrochemical degradation phenomenon in the fatigued cells during the charge/discharge cyclic tests seems to be accelerated by the accumulated mechanical damages. To examine the possibility of electrochemical reaction changes by the mechanical fatigue damages, cyclic voltammograms of the cells were obtained (Appendix A). However, it can be observed that physical stress does not evoke new phase transformations in both electrodes.

Meanwhile, the cycling performances of the batteries were very different with the initial SoC, revealing that the initial SoC of the batteries before the mechanical fatigue test can strongly affect the cell performance after the fatigue test. The SoC 70% cell exhibited stable capacity retention with a gradual decrease in capacity until the end of the cycling test, although the capacity of the SoC 70% cell was more faded than that of the cell without fatigue (Figure 4c). An SoC of 70% indicates that the graphite anode is 70% full of Li ions and the layered oxide cathode is 30% full of Li ions compared with the fully lithiated states (i.e., SoC 0% and SoC 100%). For the SoC 70% cell, both electrodes well maintained their initial state during the mechanical fatigue test and the cell capacity was well retained during the cycling test. Except for the case of the SoC 70% cell, the other cells showed drastic capacity fade. In particular, the SoC 0%, SoC 20%, and SoC 40% cells exhibited a dramatic capacity drop within the initial 20 cycles and the capacity retention was 35.0%, 49.4%, and 42.7%, respectively, for each cell. On the other hand, the SoC 100% cell displayed a different behavior. In this case, the capacity was found to exhibit a relatively gradual fading and the capacity retention at the 50th cycle was 52.8%.

In the case of SoC 0–40%, graphite anodes have 0–40% of Li in the graphite materials and the layered cathodes contain 60–100% of Li in the oxide materials. In the SoC 100% cell, the entire Li ions exist in the anode electrode. The low or high SoC implies that a large amount of Li ions existed in the cathode or anode, respectively. When the electrodes were highly lithiated during the mechanical fatigue test, it is believed that the mechanical fatigue stress causes more severe damage to the cells. Particularly, in the presence of a large amount of Li ions in the cathode (i.e., low SoC), the repetitive squeezing deformation may cause critical damage causing a sharp drop in the cell capacity during the initial cycles. This result can be interpreted as the cathode including high ratios of Li in the crystal structure being vulnerable to mechanical fatigue stress.

In conclusion, the mechanical fatigue stress affected the capacity fade of the cells in different ways in terms of the SoC. When either the cathode or anode was in a fully lithiated state (i.e., SoC 0% and 100%), the mechanical fatigue negatively affected the capacity retention of the cell during the cycle test. In particular, the rapid capacity fade was prominent in the low SoC cells. However, when both electrodes were in moderately lithiated states (i.e., SoC 70%), the cell could relatively well endure the mechanical fatigue stress. To investigate the exact mechanism of the electrode degradation, further material analyses on the anode and cathode materials after the electrochemical cycling test should be performed. It is expected that the following work can be done in the near future.

### 3.2. Effect of the Initial SoC Prior to the Mechanical Test on Electrode Resistance

To investigate the interfacial resistance changes of the electrodes in the cells based on the mechanical fatigue and cycling tests (200 cycles), EIS was used. Here, the SoC also implies the initial charge state of the cells before the fatigue tests. Figure 5 shows the equivalent electrical circuit (EEC) and the Nyquist plots of the cells under different cell conditions (i.e., the as-received state before the tests, after the fatigue test, and after the fatigue and cycling test). By fitting the EECs, the electrochemical phenomenon inside the battery can be predicted and analyzed by electrical components in the impedance response [30,31,32,33,34]. The Nyquist plots of full cells are generally assigned to two quasi-semicircles at high and mid frequencies, and a straight sloping line at low frequencies [35,36,37]. The EEC consisted of electrolyte solution resistance (R_E_), interfacial resistance of anode (R_A_), interfacial resistance of cathode (R_C_), and Warburg components (W) (Figure 5a) [35,36,37]. The anode and cathode interfacial resistance can comprise of a solid electrolyte interphase (SEI) and charge transfer resistances. In this study, the Nyquist plots are represented by two semi-circles in which the first semi-circle indicates the anode resistance (R_A_) and the second semi-circle (R_C_) indicates the cathode resistance, which was determined from the resistance characteristics of each electrode. The summed value of the three resistance terms is defined as the total resistance (R_Total_ = R_E_ + R_A_ + R_C_).

Figure 6 and Table 1 show the resistance values for all cells calculated by fitting the Nyquist plots (Figure 5). Basically, after the charge/discharge cyclic tests, the resistance of all fatigued cells was higher than that of the non-fatigued cell. It means that the induced physical damages act as the acceleration factors for electrochemical degradation. The R_Total_ values of the fatigued cells were higher than those of as-received cells and increased with decreasing the SoC (Figure 6a). In particular, the increments of R_Total_ were prominent for the SoC 0% (i.e., 9.302→2379 Ω) and 20% (i.e., 8.359→124.89 Ω) cells. After the charge/discharge cyclic tests on the fatigued cells, the R_Total_ values of the SoC 0–60% cells became extra-high. On the other hand, when the R_Total_ values of the SoC 70% (i.e., 21.44 Ω) and SoC 100% (i.e., 43.09 Ω) cells were compared with those of the SoC 0% cell (i.e., 5115 Ω), the changes in the resistance values were relatively small. The drastic increment of internal resistance implies the failure of the cells. It could be interpreted that the SoC 0% and 20% cells physically failed due to the fatigue stress, considering that the R_Total_ values were notably increased after the fatigue test. On the other hand, it is believed that the SoC 40% and 60% cells failed due to the accelerated electrochemical degradation during the cyclic test rather than physically failing after the fatigue test. The noticeable thing in the R_Total_ result is that the failure of cells predominantly tended to occur for the lower SoC cells below 60%. This implies that the low SoC cell, where the cathode is fully lithiated, is vulnerable to mechanical fatigue damages compared to the moderated and high SoC.

Between the SoC 70% cell and the SoC 100% cell, the R_Total_ values of the SoC 100% cell increased more after the charge/discharge cyclic test. The SoC 100% cell where the anode was fully lithiated was further affected by the mechanical test relative to the SoC 70% cell where the Li ions were distributed to both electrodes before the mechanical test. From these results, it can be concluded that the biased insertion of Li ions into one electrode before mechanical deformation can act as a factor that either makes it physically fail or accelerates the electrochemical degradation of the cell. The Bode plots after the fatigue test at different initial SoCs are shown in Appendix A. In the Bode plot, the SoC 0% and SoC 20% cells present different trends of the phase angle over the frequency range from 1 MHz to 0.01 Hz, confirming the highly discharged state before fatigue negatively influenced the cycling stability of the pouch-type batteries (Figure 4). These results of R_Total_ are coincident with the results of OCV in Figure 4a and capacity retention in Figure 4b,c.

Figure 6b,c shows the R_E_, R_A_, and R_C_ of all cells, respectively. The R_E_ values did not significantly change regardless of the initial SoCs and test conditions because the electrolyte remained in good condition without any influence of the tests. In the SoC 70% and 100% cells, the R_A_ and R_C_ values in each cell were similar after the fatigue and charge/discharge cyclic test. This result implies that the cathode and anode electrodes were damaged to a similar degree by the fatigue under the relatively high SoC. On the other hand, in the SoC 0–60% cells, the R_C_ significantly increased compared to the R_A_ after the fatigue test and cyclic test. This implies that the cathode electrode was more severely damaged compared to the anode electrode under the relatively low SoC. It can be concluded that the lithiated cathode electrode was most vulnerable to the physical damage by repetitive mechanical deformation. Based on the results of the OCV (Figure 4a), capacity retention (Figure 4b–d) and EIS (Figure 5 and Figure 6), the effects of mechanical fatigue on the different SoC cells are summarized in Table 2.

### 3.3. Effect of Mechanical Test Conditions

Figure 7 shows the relation between the SoH and ∆R_Total_ after the mechanical test under various conditions for the SoC 70% cell. The R_Total_ is the summation of R_E_, R_A_, and R_C_. The ∆R_Total_ indicates the increment of resistance after the fatigue and cyclic test as represented in Equation (1).
(1)$$\Delta R_{\mathrm{Total}}=\frac{R_{\mathrm{Total},\mathrm{end}}-{\;R}_{\mathrm{Total},\;\mathrm{initial}}}{R_{\mathrm{Total},\;\mathrm{initial}}}\times 100$$
where the R_Total,end_ means the R_Total_ values after the fatigue test or charge/discharge cyclic test and the R_Total,inital_ means the R_Total_ values before the tests (i.e., as-received cells). After the cycling test (200 cycles), the SoH of the non-fatigued battery was 94.1% and the resistance of 29.4% increased, as illustrated in the blue area. However, when the mechanical fatigue test under various conditions was introduced on the batteries, the SoH showed different tendencies. Regardless of the type of external force, the resistance does not change much immediately after the mechanical fatigue test. However, the mechanical effect is reflected in the cycling test. The green area illustrates the resistance change based on the mechanical fatigue mode (bending and torsion) and fatigue angle (10°, 15°, 20°, 25°, and 30°). After the cycling test, the capacity fade was highly accelerated and the resistance increased dramatically with the increase in the torsion and bending angle. It can be concluded that the mechanical fatigue test under various conditions is a stress factor that affects the cycle performance.

To investigate the effects of the torsion and bending angle, the resistance changes of the SoC 70% cells were calculated. The results after the mechanical and cycling tests based on torsion and the bending angle are given in Table 3 and Table 4. Figure 8 shows the resistance changes as the angle increases. Immediately after the torsional and bending tests, both electrodes show little change under all fatigue angles. However, the resistance values of both electrodes significantly increase after the cycling test. In the case of the torsion test, the resistance sharply increases when the angle is more than 25°. At the test angle of 30°, the cell may have been short-circuited. In the case of the bending test, the interfacial resistance of the cathode considerably increased with the increase in the angle. The change was the highest at 20°. It was believed that the cathode was more vulnerable to bending deformation than the anode, even at 70% SoC.

### 3.4. Mechanical Test Effect on the Separators

To investigate the mechanical test effect on the separators in the batteries, FE-SEM and a puncturing test were performed. The FE-SEM images of the separators are shown in Figure 9. The pristine separator shows the semi-crystalline lamellar structure consisting of organized amorphous layers (Figure 9a). For the separators in the three cells (initial SoC at 70% condition: after the cycling test, after the fatigue test, and after the cycling and fatigue test), FE-SEM images were observed. There was no remarkable difference after the tests.

To compare the mechanical properties of the separators, a puncturing test was performed on the four separators [38]. Figure 10 shows the force–displacement curves of the separators obtained from the tested cells. After the cycling test (200 cycles) without any mechanical test, the result was similar to that of the pristine separator. This indicates that the electrochemical reactions of the electrodes did not affect the mechanical properties of the separators. However, the displacement under a fixed force increased after the fatigue test, revealing that the strength of the separator degraded. Moreover, the displacement value further increased after the fatigue and cycling tests. This indicates that the subsequent cycling of the batteries had an influence on the strength of the separators. This result can be attributed to the strain-softening phenomenon [39]. It is concluded that the mechanical fatigue test on the batteries before the cycling test can affect the separators in the cells after the cycling test.

## 4. Conclusions

Pouch-type Li-Pos with different initial SoCs (0%, 20%, 40%, 60%, 70%, and 100%) were subjected to repeated torsional stress and then charge–discharge cycling of the batteries was performed 200 times. EIS analyses demonstrated the interfacial resistance changes of the anode and cathode in the cycled cells. Based on the electrochemical cycle test and EIS analysis results, the initial SoC of 70% before mechanical deformation is the best condition, in which the anode and cathode well endured the mechanical torsion test. The battery exhibited the best capacity retention and minimal resistance increase in both electrodes. This indicates that both electrodes had optimum lithiation states that were not fully lithiated states and were able to endure mechanical deformation. For the other batteries with different initial SoCs, the batteries subjected to the mechanical test showed relatively high-capacity fading. It is believed that the stress accumulated inside the batteries caused by mechanical fatigue can act as a degradation factor during cycling. As the first study to obtain insight into fatigue damage on LIBs with different initial SoCs, commercial Li-Pos were used for this study. In future research, a study on this topic would be conducted using a flexible-type battery, to more practically support the usage of LIBs in flexible electronics.

## Figures and Tables

**Figure 1 materials-15-05557-f001:**
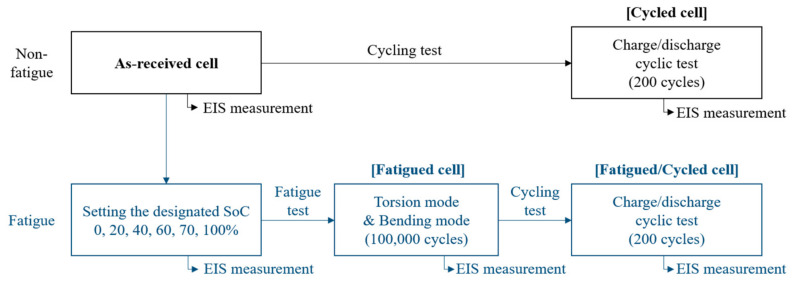
The experimental flow chart of this study.

**Figure 2 materials-15-05557-f002:**
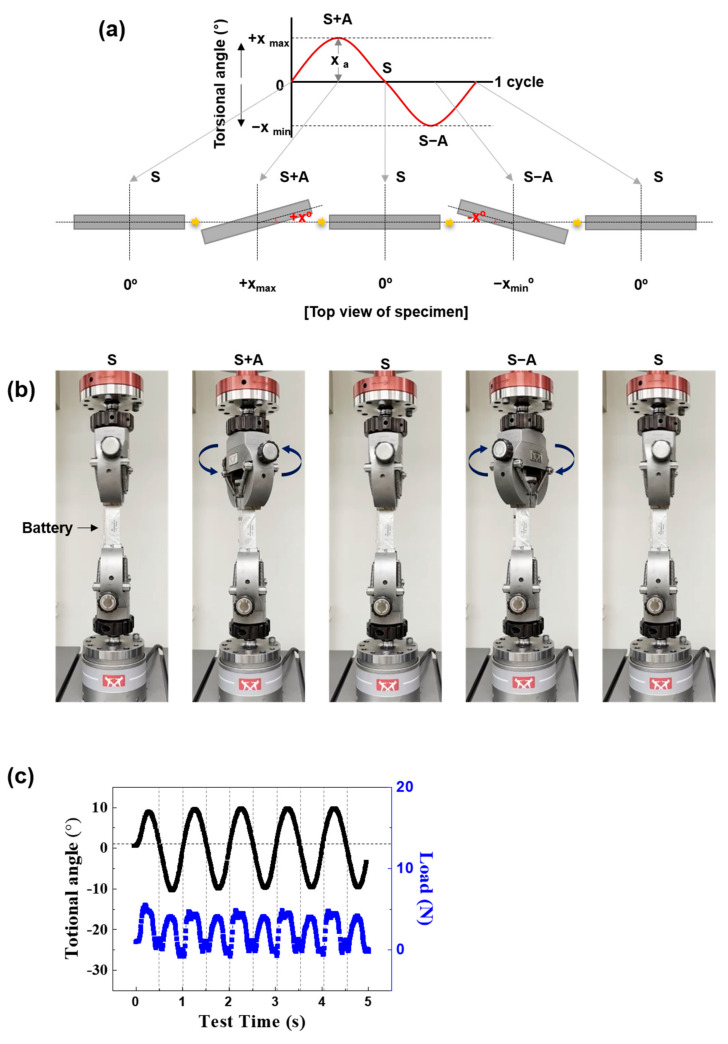
The torsional fatigue test conditions: (**a**) scheme of the phase changes in a sinusoidal wave, (**b**) torsional testing machine portraying the flexible conditions, and (**c**) fatigue load changes based on the rotation of the battery.

**Figure 3 materials-15-05557-f003:**
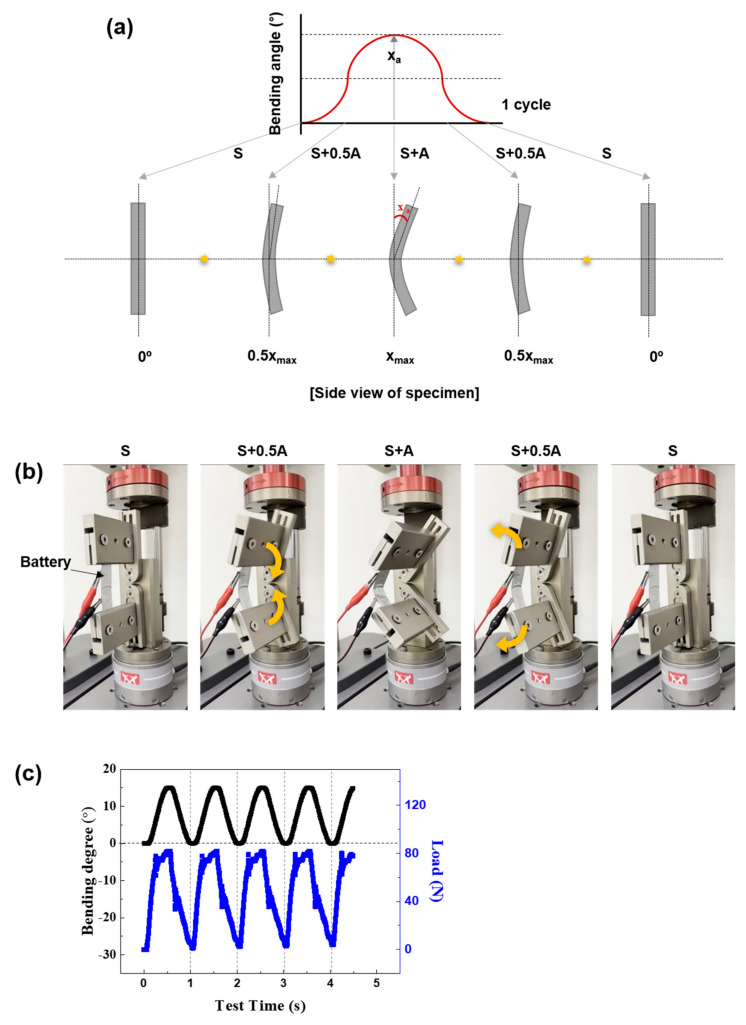
The bending fatigue test conditions: (**a**) scheme of the phase changes in a sinusoidal wave, (**b**) bending testing machine portraying the flexible conditions, and (**c**) fatigue load changes based on the rotation of the battery.

**Figure 4 materials-15-05557-f004:**
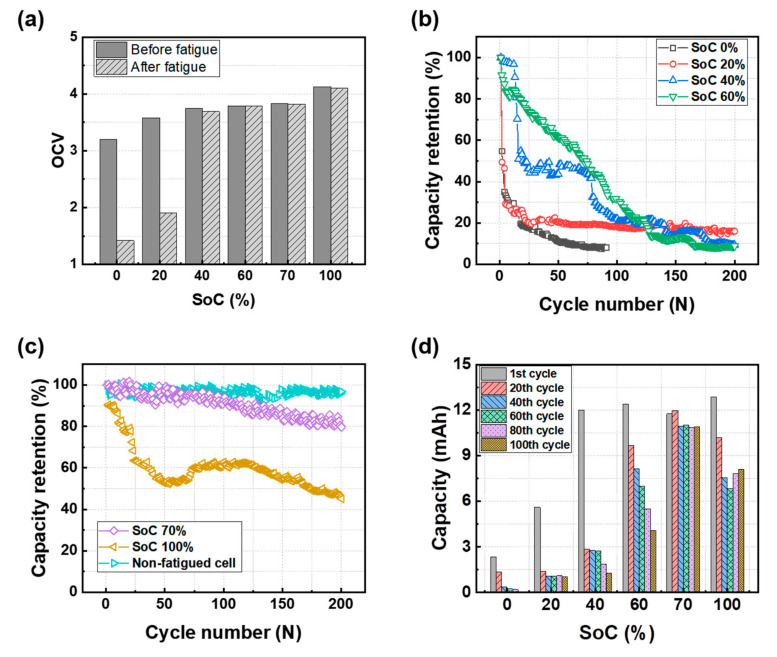
(**a**) OCV and (**b**,**c**) cycling performance of the pouch-type batteries after the mechanical fatigue test (torsion test at 10°, 100,000 cycles) with different initial SoCs; (**d**) the capacity after various cycles under different initial SoC conditions.

**Figure 5 materials-15-05557-f005:**
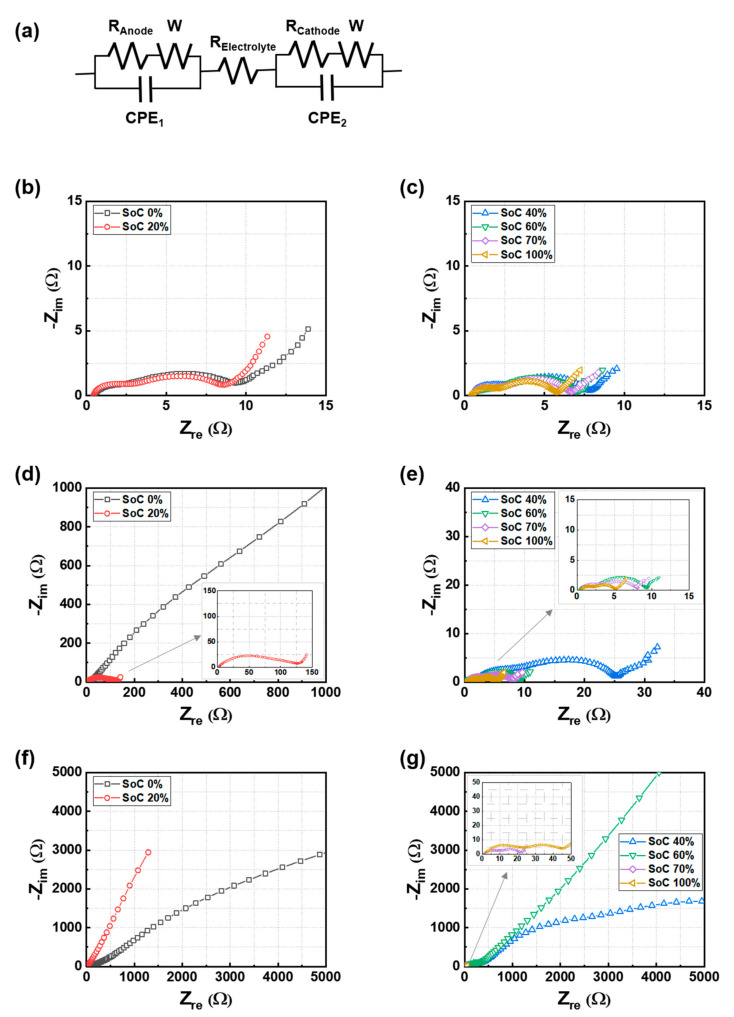
(**a**) Equivalent electrical circuit and the Nyquist plots for the different SoC cells under the conditions of (**b**,**c**) as-received, (**d**,**e**) after the fatigue test, and (**f**,**g**) after the fatigue and cycling test.

**Figure 6 materials-15-05557-f006:**
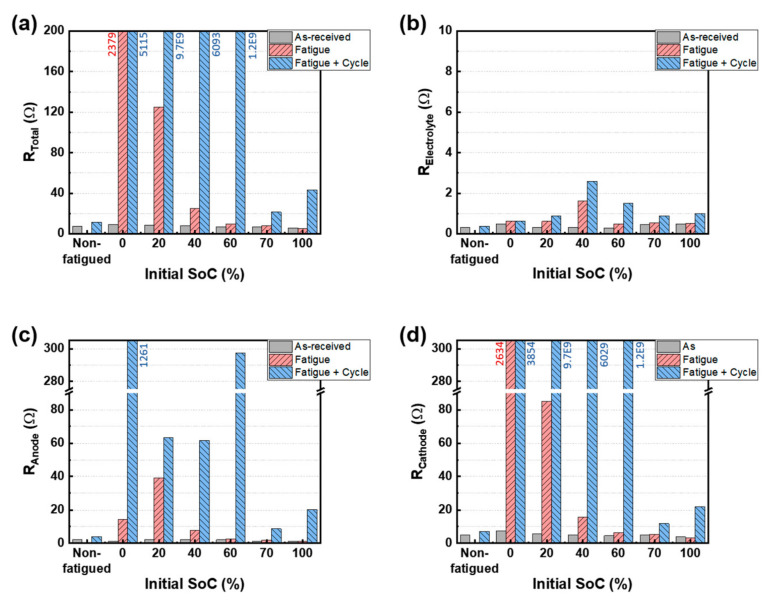
(**a**) total resistance (R_Total_), (**b**) electrolyte resistance (R_E_), (**c**) anode resistance (R_A_), and (**d**) cathode resistance (R_C_) for the different SoC cells under the conditions of as-received, after the fatigue test, and after the fatigue and cycling test.

**Figure 7 materials-15-05557-f007:**
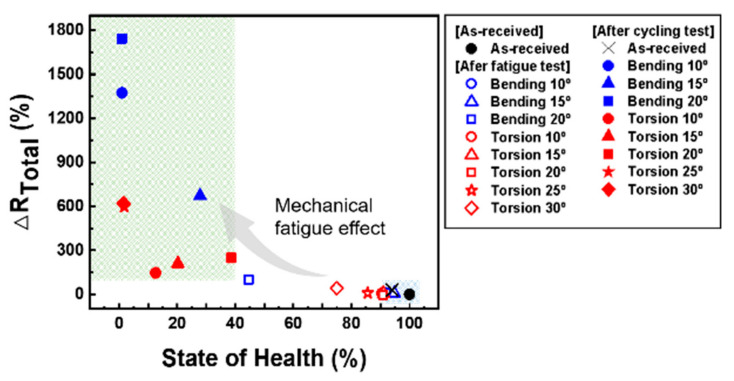
Plot of the relation between ∆R_Total_ and SoH under various test conditions.

**Figure 8 materials-15-05557-f008:**
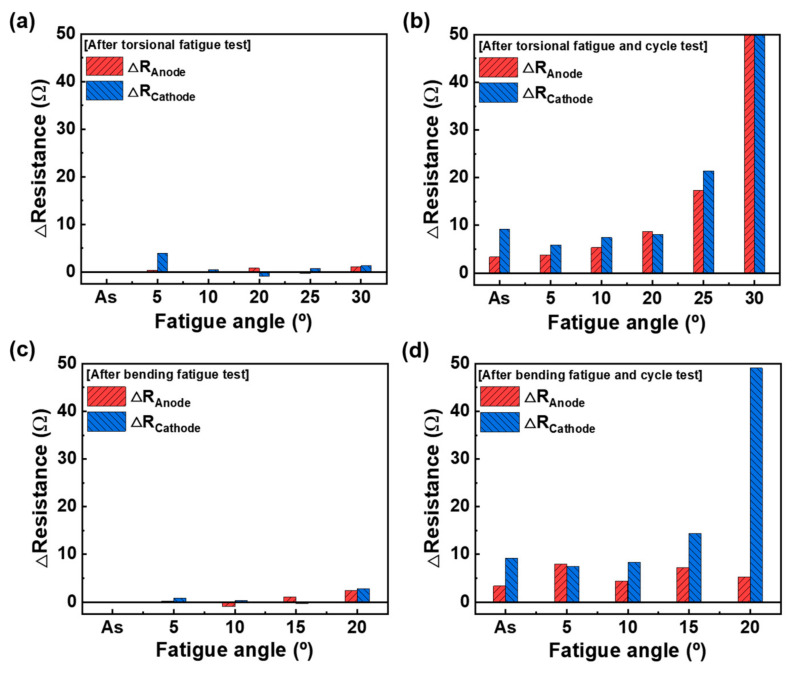
Anode and cathode resistance changes under different torsional and bending test degrees: (**a**) after the torsional fatigue test, (**b**) after the torsional fatigue and cycling test, (**c**) after the bending fatigue test, and (**d**) after the bending fatigue and cycling test.

**Figure 9 materials-15-05557-f009:**
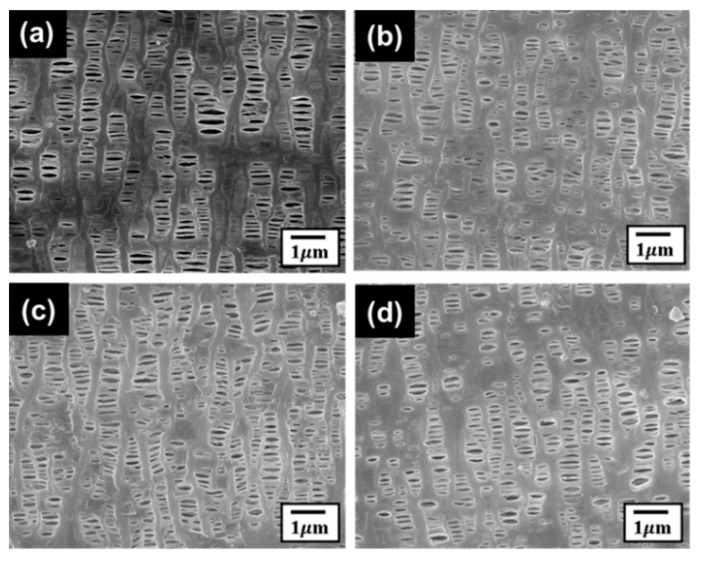
FE-SEM images of the separators under different conditions: (**a**) pristine in a fresh cell, (**b**) after the cycling test, (**c**) immediately after the fatigue test, and (**d**) after the fatigue and cycling test.

**Figure 10 materials-15-05557-f010:**
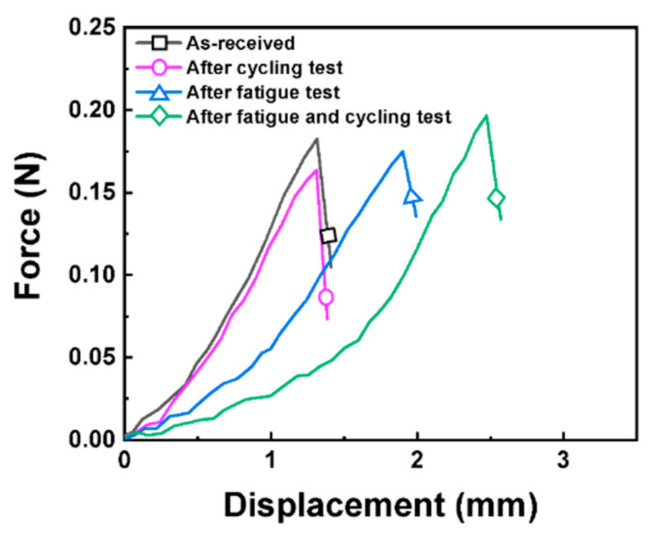
Force and displacement curves as a result of the puncturing test on the separators in the cells under various test conditions.

**Table 1 materials-15-05557-t001:** Electrolyte resistance (R_E_), anode resistance (R_A_), cathode resistance (R_C_), and total resistance (R_Total_) for the different SoC cells under the conditions of as-received, after the fatigue test, and after the fatigue and cycling test. The resistance unit is Ω.

Cell	Test Conditions	R_E_	R_A_	R_C_	R_Total_
Without Fatigue	As-received	0.323	2.202	4.857	7.382
	After cycling test	0.370	3.954	7.046	11.37
SoC 0%	As-received	0.489	1.347	7.466	9.302
	After fatigue test	0.616	14.41	2364	2379
	After fatigue and cycling test	0.624	1261	3854	5115
SoC 20%	As-received	0.309	2.377	5.673	8.359
	After fatigue test	0.610	39.16	85.12	124.89
	After fatigue and cycling test	0.882	6.315	9.7 × 10^9^	9.7 × 10^9^
SoC 40%	As-received	0.309	2.371	4.953	7.633
	After fatigue test	1.609	7.774	15.74	25.12
	After fatigue and cycling test	2.579	61.67	6029	6093
SoC 60%	As-received	0.284	2.071	4.485	6.840
	After fatigue test	0.472	2.562	6.378	9.412
	After fatigue and cycling test	1.489	297.5	1.2 × 10^9^	1.2 × 10^9^
SoC 70%	As-received	0.464	1.346	4.794	6.604
	After fatigue test	0.543	1.965	5.338	7.846
	After fatigue and cycling test	0.891	8.744	11.80	21.44
SoC 100%	As-received	0.477	1.251	3.850	5.578
	After fatigue test	0.521	1.195	3.234	4.950
	After fatigue and cycling test	0.986	20.27	21.84	43.09

**Table 2 materials-15-05557-t002:** Summarization on the effect of mechanical fatigue on the different SoC cells.

SoC	Lithiation State		Damage Levels	Mainly Damaged Part (Underlying Damage Mechanism)
	Cathode	Anode		
Low SoC	Highly lithiate	Highly delithiate	Most severe	Cathode electrode (Physical failure or accelerated electrochemical degradation)
Moderate SoC	Moderately lithiate	Moderately lithiate	Moderate	Cathode and anode electrode (Accelerated electrochemical degradation)
High SoC	Highly delithiate	Highly lithiate	Severe	

**Table 3 materials-15-05557-t003:** Change values of the electrolyte resistance (∆R_E_), anode resistance (∆R_A_), and cathode resistance (∆R_C_) after the mechanical and cycling tests under different torsional test angles.

Fatigue Angle	∆Resistance			
	∆R_E_	∆R_A_	∆R_C_	∆R_Total_
As-received	0.34	3.34	9.20	12.9
5	0.24	3.80	5.90	9.94
10	0.85	5.34	7.41	13.6
20	0.21	8.67	8.07	17.0
25	2.96	17.3	21.4	41.7
30	0.68	260	102,000	102,260

**Table 4 materials-15-05557-t004:** Change values of the electrolyte resistance (∆R_E_), anode resistance (∆R_A_), and cathode resistance (∆R_C_) after the mechanical and cycling tests under different bending test angles.

Fatigue Angle	∆Resistance			
	∆R_E_	∆R_A_	∆R_C_	∆R_Total_
As-received	0.34	3.34	9.20	12.9
5	0.71	8.01	7.51	16.2
10	0.27	4.37	8.33	13.0
15	1.01	7.19	14.4	22.6
20	4.01	5.20	49.7	58.9

## Data Availability

Not applicable.

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
