# Peer review of "Influence of Mechanical Fatigue at Different States of Charge on Pouch-Type Li-Ion Batteries"

_materials, 2022, doi:10.3390/ma15165557_

Round 1
Reviewer 1 Report
In this work, Kim and coworkers tested pouch-type LIBs with different initial SoCs. The LIBs were subjected to repeated torsional stress for 100k cycles, and then electrochemically cycled 200 times. Moreover, through a variety of characterizations, the identification and comparison of the optimum SoC for wearable device applications were carried out. The results showed that there were great differences in the mechanical stability to torsional stress with different SoCs. This is a meaningful study, and I agree to publish it on the "materials" after a very minor modifications.
1. In line 44, the author mentioned "layered transition metal oxides (LiNixCoyMnzO2, x+y+z = 2)". I think x+y+z should equal 1, not 2.
2. Most fonts in the figures appear to be TOO SMALL. Please consider to revise the following:
- Figure 1C axis number.
- Figure 2a legend, axis number. Please also consider reducing the number of data points to enhance the readability.
- Figure 4 legend.
- etc.
Because the journal "Materials" displays the figures as a full-width only, you can consider to show them in more expanded width than before.
3. In Figure 2b, is there any specific reason to show capacity vs. SoC by using a line graph? A line graph is generally used to show the y-axis changes over the x-axis. I think the way this graph drawn is inappropriate because, for example, connecting the points between SoC 0, 20, 40, 60, 80, and 100 of 1st cycles doesn't have any scientific meanings. There will be a better way to convey the authors intension.
Author Response
Thank you for the valuables comments.
We attached the cover letter and response to your comment.
Best regards,
Jaiwon Byeon

Reviewer 2 Report
The article titled “Influence of mechanical fatigue at different states of charge on pouch-type Li-ion batteries” by Kim et al. describes the analysis of fatigue tests on pouch cells at different SoC. The research provides interesting perspective of the performance of LIBs under mechanical fatigue; however, further information is needed to better evaluate the merit of this work.
1. In the introduction part, the authors claim the purpose of this work is to support the usage of LIBs in flexible electronics. However, it is barely feasible to use commercial pouch cells (with multiple layers of cathodes, anodes and electrolyte, and the pouch material is laminated Al/polymer foil) for flexible electronics which is usually using polymer thin films as substrates. Please provide further justification for this work.
2. Were any formation cycles performed to the tested cells prior to discharging to designated SoC?
3. For the torsion test (Figure 1b), is that correct that one of the two fixtures is performing the torsion as shown in Figure 1a, and the other one is fixed? For the bending test, it would be great to further describe the testing condition.
4. For results in Figure 2a, how many samples were tested for each initial SoC? How was the variation among different samples?
5. Page 4, line 152, “The cathode, including the Li ions, may have been further affected by mechanical deformation, causing a sharp drop in the cell capacity during the initial cycles.” What exactly in the cathode is being affected? It is hard to understand the mechanism why 70% SoC cells had the best capacity retention.
6. For the EIS results, why is the Nyquist plots interpreted into the resistances of electrolyte, cathode and anode? Please explain the underlying physics of these resistances (ion transportation, electron transportation)? Why are the cathode and anode resistances at different frequencies? Please further elaborate. Before understanding this, it is hard to understand the all following results on ΔR. What is the meaning to analyze ΔRtotal?
Author Response

(The authors gave the same response as above.)

Round 2
Reviewer 2 Report
I am happy with the authors' reply and revised manuscript.
I recommend accept it. Thank you!